# Uncertainty-aware Distributional Offline Reinforcement Learning

## Abstract

Offline reinforcement learning (RL) presents distinct challenges as it relies solely on observational data. A central concern in this context is ensuring the safety of the learned policy by quantifying uncertainties associated with various actions and environmental stochasticity. Traditional approaches primarily emphasize mitigating epistemic uncertainty by learning risk-averse policies, often overlooking environmental stochasticity. In this study, we propose an uncertainty-aware distributional offline RL method to simultaneously address both epistemic uncertainty and environmental stochasticity. We propose a model-free offline RL algorithm capable of learning risk-averse policies and characterizing the entire distribution of discounted cumulative rewards, as opposed to merely maximizing the expected value of accumulated discounted returns. Our method is rigorously evaluated through comprehensive experiments in both risk-sensitive and risk-neutral benchmarks, demonstrating its superior performance.

## 1 Introduction

In the domain of safety-critical applications, the practicality of implementing online RL is constrained due to the imperative for extensive random exploration, which may incur potential hazards or substantial costs. In response to this predicament, offline RL (Levine et al., 2020) endeavors to discern the most optimal policy based on pre-collected data. In safety-critical settings, there is a pronounced emphasis on cultivating uncertainty-averse decision-making while factoring in the inherent variability in performance. Regrettably, contemporary research in offline RL predominantly fixates on objectives associated with expected value, thereby neglecting the consideration of performance variability across distinct episodes.

There are two different types of uncertainties in offline RL: epistemic and aleatoric. Epistemic uncertainty is caused by model itself and aleatoric uncertainty is caused by the environment (i.e., pre-collected data). Previous works address the epistemic uncertainty by using the risk-averse offline RL. The common approach is to use variational autoencoders (VAEs) to reconstruct the behavior policy, which can also help reduce bootstrapping errors (Urpí et al., 2021; Lyu et al., 2022). However, these approaches heavily rely on imitation learning to align the learned policy with the empirical behavior policy, potentially leading to sub-optimal performance due to the presence of sub-optimal trajectories and the risk-neutral nature of the dataset (Ma et al., 2021). Offline RL leverages large datasets, but incorporating data from multiple behavioral policies presents challenges in maintaining fidelity to a high-quality policy (Ma et al., 2021; Wang et al., 2023; Kumar et al., 2022). Another alternative involves refining behavior policies (Ma et al., 2021), yet it lacks versatility for different environments or tasks. Additionally, the collected offline trajectories consist of a mixture of trajectories from various behavior policies. VAE-based methods struggle to distinguish these trajectories, resulting in sub-optimal performance. Furthermore, recent findings highlight VAE's constraints in modeling complicated distributions (Pearce et al., 2023; Wang et al., 2023).

In terms of the aleatoric uncertainty, it is caused by the environment stochasticity. It will affect the policy learning via the accumulated discounted returns. Previous works (Bellemare et al., 2017; Barth-Maron et al., 2018) propose to use the distributional offline RL to formulate the randomness where the distribution of the discounted cumulative reward is used instead of maximizing the expectation of accumulated discounted returns.

In this study, we address the challenge of handling both uncertainties simultaneously within the context of risk-averse offline RL. As discussed before, existing epistemic-aware methods (i.e., risk-averse offline RL) have the limitation of expressiveness caused by VAE. Inspired by the recent advancement of the diffusion model, which has a strong expressiveness when imitating human behaviors (Pearce et al., 2023), we design a novel off-policy RL algorithm method named Uncertainty-aware offline Distributional Actor-Critic (UDAC). UDAC leverages the diffusion model as a key component in behavior policy modeling, offering several advantages: i). *Elimination of Manual Behavior Policy Specification*: UDAC eliminates the need for manually defined behavior policies, making it adaptable to a wide range of risk-sensitive RL tasks. ii). *Accurate Modeling of the behavior policy*: UDAC enhances the precision of modeling the full distribution of behavior policy through the diffusion model, improving robustness in the face of environmental stochasticity. iii). *Incorporation of Perturbation Model*: UDAC goes beyond simple imitation learning by incorporating a perturbation model tailored to meet the requirements of risk-sensitive settings.

We conducted extensive experiments in different benchmarks covering both risk-sensitive and risk-neutral scenarios: risk-sensitive D4RL, risky robot navigation, and risk-neutral D4RL. Our results demonstrate that UDAC outperforms most baselines in risk-sensitive D4RL and risky robot navigation. Moreover, UDAC achieves comparable performance to existing state-of-the-art methods in risk-neutral D4RL.[1]

## 2 RELATED WORKS

Pearce et al. (2023) suggest that the imitation of human behaviors can be improved through the use of expressive and stable diffusion models. Janner et al. (2022) employ a diffusion model as a trajectory generator in their Diffuser approach, where a full trajectory of state-action pairs forms a single sample for the diffusion model. Singh et al. (2020) utilize a distributional critic in their algorithm, but it is restricted to the CVaR and limited to the online reinforcement learning (RL) setting. Furthermore, their use of a sample-based distributional critic renders the computation of the CVaR inefficient. To enhance the computational efficiency of the CVaR, Urpí et al. (2021) extend the algorithm to the offline setting. Ma et al. (2021) replace normal Q-learning with conservation offline Q-learning and an additional constant to penalize out-of-distribution (OOD) behaviors. However, Lyu et al. (2022) observe that the introduced extra constant is excessively severe and propose a more gentle penalized mechanism that uses a piecewise function to control when should apply the extra penalization.

**Differences between existing works.** There are two similar works that also apply the diffusion model in offline reinforcement learning: Diffusion-QL (Wang et al., 2023) and Diffuser (Janner et al., 2022). Diffuser (Janner et al., 2022) is from the model-based trajectory-planning perspective, while Diffusion-QL (Wang et al., 2023) is from the offline model-free policy-optimization perspective. But our work is from the risk-sensitive perspective and extends it into the distributional offline reinforcement learning settings.

## 3 BACKGROUND

In this work, a Markov Decision Process (MDP) is considered with states $s \in \mathcal{S}$ and actions $a \in \mathcal{A}$ that may be continuous. It includes a transition probability $P(\cdot|s, a) : \mathcal{S} \times \mathcal{A} \times \mathcal{S} \to [0, 1]$, reward function $R(\cdot|s, a) : \mathcal{S} \times \mathcal{A} \to \mathbb{R}$, and a discount factor $\gamma \in [0, 1)$. A policy $\pi(\cdot|s)$ is defined as a mapping from states to the distribution of actions. RL aims to learn $\pi(\cdot|s)$ such that the expected cumulative long-term rewards $\mathbb{E}_{s_0, a_t \sim \pi(\cdot|s_t), s_{t+1} \sim P(\cdot|s_t, a_t)}[\sum_{t=0}^{\infty} \gamma^t R(\cdot|s_t, a_t)]$ are maximized. The state-action function $Q_\pi(s, a)$ measures the discounted return starting from state $s$ and action $a$ under policy $\pi$. We assume the reward function $R(\cdot|s, a)$ is bounded, i.e., $|R(\cdot|s, a)| \leq r_{max}$. Now, consider the policy $\pi(\cdot|s)$, the Bellman operator $\mathcal{T}^\pi : \mathbb{R}^{|\mathcal{S}||\mathcal{A}|} \to \mathbb{R}^{|\mathcal{S}||\mathcal{A}|}$ is introduced by Riedmiller (2005) to update the corresponding $Q$ value,

$$\mathcal{T}^\pi Q(s, a) := \mathbb{E}[R(s, a)] + \gamma \mathbb{E}_{P(s'|s, a)\pi(a'|s')}[Q(s', a')]. \tag{1}$$

In offline RL, the agent can only access a fixed dataset $\mathcal{D} := \{(s, a, r, s')\}$, where $r \sim R(\cdot|s, a), s' \sim P(\cdot|s, a)$. $\mathcal{D}$ comes from the behavior policies. We use $\pi_b$ to represent the behavior policy, and $(s, a, r, s') \sim \mathcal{D}$ represents the uniformly random sampling.

---

[1]Code is available at `https://anonymous.4open.science/r/UDAC-0241`.

In distributional RL (Bellemare et al., 2017), the aim is to learn the distribution of discounted cumulative rewards, represented by the random variable $Z^\pi(s,a) = \sum_{t=0}^{\infty} \gamma^t R(\cdot|s_t, a_t)$. Similar to the $Q$ function and Bellman operator mentioned earlier, we can define a distributional Bellman operator as follows,

$$\mathcal{T}^\pi Z(s,a) :=_D R(s,a) + \gamma Z(s', a'). \quad s' \sim P(\cdot|s,a), a' \in \pi(\cdot|s'). \quad (2)$$

The symbol $:=_D$ is used to signify equality in distribution. Thus, $Z^\pi$ can be obtained by applying the distributional Bellman operator $\mathcal{T}^\pi$ iteratively to the initial distribution $\mathcal{Z}$. In offline settings, $\mathcal{T}^\pi$ can be approximated by $\hat{\mathcal{T}}^\pi$ using $\mathcal{D}$. Then, we can compute $Z^\pi$ by starting from an arbitrary $\hat{Z}^0$, and iteratively minimizing the p-Wasserstein distance between $Z$ and $\hat{\mathcal{T}}^\pi \hat{Z}^\pi$ (To simplify, we will use the $W_p(\cdot)$ to represent the p-Wasserstein distance). And the measure metric $\overline{d_p}$ over value distributions is defined as,

$$\overline{d_p}(Z_1, Z_2) = \sup_{s,a} W_p(Z_1(s,a), Z_2(s,a)), Z_1, Z_2 \in \mathcal{Z}.$$

## 4 METHODOLOGY

Our proposed method, UDAC, is based on the quantile-based distributional RL framework, comprising three parts: quantile value network, policy network $\pi_\theta(s)$ and the behavior policy network $\pi_{db}$. The overall framework for UDAC is outlined in Section 4.1. The loss of the actor is defined as the risk-distortion operator[2] applied to the learned return distribution and optimized through gradient-based methods. With this actor-critic setup, a risk-averse criterion can be optimized. However, in the offline setting, controlling the bootstrapping error is crucial. To address this, a diffusion policy is incorporated in the actor to learn a generative model of the behavior policy, which is described in Section 4.2. Finally, all components are combined in Section 4.3 to implement the algorithm for a given risk distortion $\mathbb{D}$.

### 4.1 OVERVIEW OF UDAC

In the critic network of UDAC, our aim is policy evaluation, and we rely on Equation (2) to achieve this objective. Following the approaches of Dabney et al. (2018a); Urpí et al. (2021); Ma et al. (2021), we implicitly represent the return through its quantile function, as demonstrated by Dabney et al. (2018a). In order to optimize the $Z$, we utilise a neural network with a learnable parameter $\theta$ to represent the quantile function. We express such implicit quantile function as $F_Z^{-1}(s,a;\tau)$ for $Z$, where $\tau \in [0,1]$ is the quantile level. Notice that, the original quantile function can only be used for discrete action. We can extend it into a continuous setting by considering all $s, a$, and $\tau$ as the inputs and only the quantile value as the output. To learn the $\theta$, the common strategy is the fitted distributional evaluation using a quantile Huber-loss $\mathcal{L}_\mathcal{K}$ Huber (1992). Our optimizing objective for the critic network is,

$$\mathcal{L}(\hat{\mathcal{T}}^\pi \hat{Z}^\pi, Z) = W_p(\hat{\mathcal{T}}^\pi \hat{Z}^\pi, Z) \simeq \mathcal{L}_\mathcal{K}(\delta; \tau), \text{ with } \delta = r + \gamma F_{Z'}^{-1}(s', a'; \tau') - F_Z^{-1}(s,a;\tau), \quad (3)$$

where $(s,a,r,s') \sim \mathcal{D}$ and $a' \sim \pi(\cdot|s')$. The $\delta$ is also known as the distributional TD error (Dabney et al., 2018b). And the quantile Huber-loss $\mathcal{L}_\mathcal{K}$ is represented as,

$$\mathcal{L}_\mathcal{K}(\delta; \tau) = \begin{cases} |1_{\delta<0} - \tau| \cdot \delta^2/(2\mathcal{K}) & \text{if } |\delta| < \mathcal{K}, \\ |1_{\delta<0} - \tau| \cdot (\delta - \mathcal{K}/2) & \text{otherwise} \end{cases} \quad (4)$$

where $\mathcal{K}$ is the threshold. Notice that there is a hyper-parameter $\tau$ which may affect our objective function. Here, we use a simple strategy to control the $\tau$: Uniformly random sampling $\tau \sim U(0,1)$. It is worth mentioning that there is an alternative way for controlling $\tau$, which is a quantile level proposal network $P_\psi(s,a)$ Yang et al. (2019). To simplify the overall model and reduce the number of hyperparameters, we just use the simplified random sampling approach and leave the quantile proposal network for future work. The overall loss function for the critic is

$$\mathbb{E}_{(s,a,r,s')\sim\mathcal{D},a'\sim\pi(\cdot|s')} \left[ \frac{1}{N*K} \sum_{i=0}^{N} \sum_{j=0}^{K} \mathcal{L}_\mathcal{K}(\delta_{\tau_i, \tau_j'}; \tau_i) \right], \quad (5)$$

---

[2]risk-distortion operator is a class of distortion function that also considers the risk Wang (2000).

where $N, K$ are the numbers of quantile levels. In the risk-sensitive setting, deterministic policies are favoured over stochastic policies due to the latter's potential to introduce additional randomness, as noted by Pratt (1978). Moreover, in offline settings, there is no benefit to exploration commonly associated with stochastic policies. Therefore, the focus is on parameterized deterministic policies, represented by $\pi_\phi(s) : \mathcal{S} \to \mathcal{A}$. Given the risk-distortion function $\mathbb{D}$, the loss function of actor is written as,

$$-\mathbb{E}_{s \sim \rho_b}[\mathbb{D}(F_Z^{-1}(s, \pi_\phi(s); \tau))] \tag{6}$$

with the marginal state distribution $\rho_b$ from the behavior policy. To optimize it, we can backpropagate it through the learned critic at the state that sampled from the behavior policy. It is equivalent to maximising risk-averse performance. Then, combining the above process with the SAC (Haarnoja et al., 2018) leads to our novel UDAC model.

## 4.2 GENERALIZED BEHAVIOR POLICY MODELING

We separate the actor into two components: an imitation component (Fujimoto et al., 2019) and a conditionally deterministic perturbation model $\xi$,

$$\pi_\phi(s) = \lambda \xi(\cdot|s, \beta) + \beta, \beta \sim \pi_b, \tag{7}$$

where $\xi(\cdot|s, \beta)$ can be optimized by using the actor loss function(i.e., $-\mathbb{E}_{s \sim \rho_b}[\mathbb{D}(F_Z^{-1}(s, a; \tau))]$) that was defined previously, and $\beta$ is the action that sampled from behavior policy $\pi_b$, $\lambda$ is a scale that used to control the magnitude of the perturbation. As aforementioned, we prefer the deterministic policy over the stochastic policy because of the randomness. However, the bootstrapping error will appear on the offline RL setting. Existing works (Urpí et al., 2021; Wu et al., 2019; Ma et al., 2021; Lyu et al., 2022) use policy regularization to regularize how far the policy can deviate from the behavior policy. But the policy regularization will adversely affect the policy improvement by limiting the exploration space, and only suboptimal actions will be observed. Moreover, the fixed amount of data from the offline dataset may not be generated by a single, high-quality behaviour policy and thus affect the imitation learning performance (i.e., the logged data may come from different behavior policies). Hence, behavioral cloning (Pomerleau, 1991) would not be the best choice. Moreover, behavioral cloning suffers from mode-collapse, which will affect actor's optimization. Here, we attempt to use the generative method to model the behaviour policy from the observed dataset. The most popular approach is the cVAE which leverages the capability of the auto-encoder to reconstruct the behavior policy. But the behavior policies on the offline dataset are normally collected from different agents and may have different modalities. It implies that some behavior policies may be sub-optimal. In this work, we treat these sub-optimal policies as noisy policies. Inspired by recent work of denoising in computer vision related tasks (Ho et al., 2020), we employ the diffusion model instead of cVAE to model the behavior policy.

We represent the behavior policy by using the reverse process of a conditional diffusion model, which is denoted by the parameter $\psi$, as follows,

$$\pi(a|s) = p(a^{0:N}|s) = \mathcal{N}(a^N; 0, I) \sum_{i=1}^{N} p(a^{i-1}|a^i, s), \tag{8}$$

where $p(a^{i-1}|a^i, s)$ can be represented as a Gaussian distribution $\mathcal{N}(a^{i-1}; \mu_\psi(a^i, s, i), \sum_\psi(a^i, s, i))$, where $\sum_\psi(a^i, s, i)$ is a covariance matrix. To parameterize $p(a^{i-1}|a^i, s)$ as a noise prediction model, we fix the covariance matrix as $\sum_\psi(a^i, s, i) = \beta_i I$ and construct the mean as,

$$\mu_\psi(a^i, s, i) = \frac{1}{\sqrt{\alpha_i}} \left( a^i - \frac{\beta_i}{\sqrt{1 - \overline{\alpha}_i}} \epsilon_\psi(a^i, s, i) \right). \tag{9}$$

To obtain a sample, we begin by sampling $a^N$ from a normal distribution with mean 0 and identity covariance matrix, $\mathcal{N}(0, I)$. We then sample from the reverse diffusion chain, which is parameterized by $\psi$, to complete the process,

$$a^{i-1}|a^i = \frac{a^i}{\sqrt{\alpha_i}} - \frac{\beta_i}{\alpha_i \sqrt{1 - \overline{\alpha}_i}} \epsilon_\psi(a^i, s, i) + \sqrt{\beta_i}\epsilon, \epsilon \sim \mathcal{N}(0, I), \text{ for } i = N, \cdots, 1. \tag{10}$$

Following the approach of Ho et al. (2020), we set $\epsilon$ to 0 when $i = 1$ to improve the sampling quality. The objective function used to train the conditional $\epsilon$-model is,

$$\mathcal{L}_d(\psi) = \mathbb{E}_{i\sim\mathcal{U},\epsilon\sim\mathcal{N}(0,I),(s,a)\sim\mathcal{D}}\big[\|\epsilon - \epsilon_\psi(\sqrt{\overline{\alpha}_i}a + \sqrt{1 - \overline{\alpha}_i}\epsilon, s, i)\|^2\big], \tag{11}$$

where $\mathcal{U}$ is a uniform distribution over the discrete set, denoted as $\{1, \cdots, N\}$. This diffusion model loss $\mathcal{L}_d(\psi)$ aims to learn from the behavior policy $\pi_b$ instead of cloning it. $\mathcal{L}_d(\psi)$ can be optimized by sampling from single diffusion step $i$ for each data point. However, it will be affected significantly by $N$, which is known as the bottleneck of the diffusion (Kong & Ping, 2021). We add this term to the actor loss and construct the final loss function. We then replace the term $\pi_b$ in the Equation (7) with $d_\psi(s, a)$ (i.e., the diffusion behavior policy representation).

### 4.3 TRAINING ALGORITHM

By combing all of the components mentioned in previous sections, the overall training algorithm can be described in Algorithm 1. The critic seeks to learn the return distribution, while the diffu-

---

**Algorithm 1:** UDAC with $U(0, 1)$

---

1 Initialize critic parameter $\theta$, actor parameter $\phi$, perturbation model $\xi$, quantile huber loss threshold $\mathcal{K}$, number of generated quantiles $N, K$, diffusion parameter $\psi$, distortion $\mathbb{D}$, learning rate $\eta$;
2 Sample quantile $\tau_i, \tau_j$, where $i = \{0, \cdots, N\}, j = \{0, \cdots, K\}$ from $\mathcal{U}(0, 1)$;
3 **for** $t = 1, \cdots$ **do**
4     Sample $B$ transitions $(s, a, r, s')$ from the offline dataset;
5     Compute the $\delta$ based on Equation (3);
6     Sample $\beta$ from $d_\psi(s, a)$ and compute the policy $\pi_\phi$ based on Equation (7);
7     Compute critic loss by using Equation (5);
8     Compute the loss for $\xi$ by using Equation (6);
9     Compute the diffusion loss by using Equation (11);
10     Compute the loss of actor by adding the loss of $\xi$ and the diffusion model;
11     Update $\theta, \phi, \psi$ accordingly;
12 **end**

---

sion model aims to comprehend the behavioral policy, serving as a baseline action for the actor to implement risk-averse perturbations. As aforementioned, the distortion operator $\mathbb{D}$ is required. It can be any distorted expectation objective, such as CVaR, Wang, CPW etc. In this work, we focus on $\text{CVaR}_{\alpha=0.1}$, which is the majority distorted measure used in recent literature (Rigter et al., 2022; Urpí et al., 2021; Ma et al., 2021).

## 5 EXPERIMENTS

In this section, we show that UDAC achieves state-of-the-art results on risk-sensitive offline RL tasks, including risky robot navigation and D4RL, and comparable performance with SOTAs in risk-neutral offline RL tasks.

### 5.1 RISK-SENSITIVE D4RL

**Tasks** Firstly, we test the performance of the proposed UDAC in risk-sensitive D4RL environments. Hence, we turn our attention to stochastic D4RL (Urpí et al., 2021). The original D4RL benchmark (Fu et al., 2020) comprises datasets obtained from SAC agents that exhibit varying degrees of performance (Random, Medium, and Expert) on the Hopper, Walker2d, and HalfCheetah in MuJoCo environments (Todorov et al., 2012). Stochastic D4RL modifies the rewards to reflect the stochastic damage caused to the robot due to unnatural gaits or high velocities; please refer to Appendix D.1 for details. The Expert dataset consists of rollouts generated by a fixed SAC agent trained to convergence; the Medium dataset is constructed similarly, but the agent is trained to attain only 50% of the expert agent's return. We have selected several state-of-the-art risk-sensitive offline RL algorithms as well as risk-neutral algorithms. It includes: O-RAAC (Urpí et al., 2021), CODAC (Ma

et al., 2021), O-WCPG (Tang et al., 2020), Diffusion-QL (Wang et al., 2023), BEAR (Kumar et al., 2019) and CQL (Kumar et al., 2020). Note that, the O-WCPG is not originally designed for offline settings. We follow the same procedure as described in Urpí et al. (2021) to transform it into an offline algorithm. Among them, ORAAC, CODAC and O-WCPG are risk-sensitive algorithms, Diffusion-QL, BEAR and CQL are risk-natural. Moreover, ORAAC and CODAC are distributional RL algorithms.

**Results** In Table 1, we present the mean and $CVaR_{0.1}$ returns on test episodes, along with the duration for each approach. These results are averaged over 5 random seeds. Additionally, we report the results on the median dataset in Appendix D.3, where UDAC also achieves the strongest performance. As shown, UDAC outperforms the baselines on most datasets. It is worth noting that CODAC contains two different variants that optimize different objectives. In this task, we select CODAC-C as the baseline since it targets risk-sensitive situations. Furthermore, we observe that the performance of CQL varies significantly across datasets. Under the risk-sensitive setting, it performs better than two other risk-sensitive algorithms, O-RAAC and TeCODAC, in terms of $CVaR_{0.1}$. One possible explanation for this observation is that CQL learns the full distribution, which helps to stabilize the training and thus yields a better $CVaR_{0.1}$ than O-RAAC and CODAC.

Table 1: Performance of the UDAC algorithm on risk-sensitive offline D4RL environments using medium and expert datasets

| | Algorithm | Medium | | | Expert | | |
|---|---|---|---|---|---|---|---|
| | | $CVaR_{0.1}$ | Mean | Duration | $CVaR_{0.1}$ | Mean | Duration |
| Half-Cheetah | O-RAAC | 214(36) | 331(30) | 200(0) | 595(191) | 1180(78) | 200(0) |
| | CODAC | -41(17) | 338(25) | 200(0) | 687(152) | 1255(101) | 200(0) |
| | O-WCPG | 76(14) | 316(23) | 200(0) | 248(232) | 905(107) | 200(0) |
| | Diffusion-QL | 84(32) | 329(20) | 200(0) | 542(102) | 728(104) | 200(0) |
| | BEAR | 15(30) | 312(20) | 200(0) | 44(20) | 557(15) | 200(0) |
| | CQL | -15(17) | 33(36) | 200(0) | -207(47) | -75(22.6) | 200(0) |
| | Ours(UDAC) | 276(29) | 417(18) | 200(0) | 732(104) | 1352(100) | 200(0) |
| Walker-2D | O-RAAC | 663(124) | 1134(20) | 397(18) | 1172(71) | 2006(56) | 432(11) |
| | CODAC | 1159(57) | 1537(78) | 403(19) | 1298(98) | 2102(102) | 402(18) |
| | O-WCPG | -15(41) | 283(37) | 185(12) | 362(33) | 1372(160) | 301(31) |
| | Diffusion-QL | 31(10) | 273(20) | 230(9) | 621(49) | 1202(63) | 365(12) |
| | BEAR | 517(66) | 1318(31) | 400(8) | 1017(49) | 1783(32) | 407(4) |
| | CQL | 1244(128) | 1524(99) | 405(55) | 1301(78) | 2018(65) | 404(43) |
| | Ours(UDAC) | 1368(89) | 1602(85) | 406(24) | 1398(67) | 2198(78) | 462(20) |
| Hopper | O-RAAC | 1416(28) | 1482(4) | 499(1) | 980(28) | 1385(33) | 494(6) |
| | CODAC | 976(30) | 1014(16) | 486(5) | 990(19) | 1398(29) | 490(12) |
| | O-WCPG | -87(25) | 69(8) | 100(0) | 720(34) | 898(12) | 301(1) |
| | Diffusion-QL | 981(28) | 1001(11) | 339(3) | 406(31) | 583(18) | 262(3) |
| | BEAR | 1252(47) | 1575(8) | 481(2) | 852(30) | 1180(12) | 431(4) |
| | CQL | 1344(40) | 1524(20) | 478(9) | 1289(30) | 1402(30) | 495(5) |
| | Ours(UDAC) | 1502(19) | 1602(18) | 480(6) | 1302(28) | 1502(27) | 494(8) |

## 5.2 RISKY ROBOT NAVIGATION

**Tasks** We now aim to test the risk-averse performance of our proposed method. Following the experimental setup detailed in Ma et al. (2021), we will conduct experiments on two challenging environments, namely Risky Ant and Risky PointMass. In these environments, an Ant robot must navigate from a randomly assigned initial state to the goal state as quickly as possible. However, the path between the initial and goal states includes a risky area, which incurs a high cost upon passing through. While a risk-neutral agent may pass through this area, a risk-sensitive agent should avoid it. For further information on the experimental setup, including details on the environments and datasets, refer to D.1. We perform the comparison with the same baselines as the previous section. We present a detailed comparison between CODAC and ORAAC, two state-of-the-art distributional risk-sensitive offline RL algorithms, as well as other risk-neutral algorithms. Our discussion covers a detailed analysis of their respective performance, strengths, and weaknesses.

**Results** The effectiveness of each approach is evaluated by running 100 test episodes and measuring various metrics, including the mean, median, and $CVaR_{0.1}$ returns, as well as the total number of violations (i.e., the time steps spent inside the risky region). To obtain reliable estimates, these metrics are averaged over 5 random seeds. Moreover, we also provide a visualization of the evaluation trajectories on the Risky Ant environment on Figure 1, and we can see that the CODAC and ORAAC do have some risk-averse action, while UDAC successfully avoids the majority of the risk area. We also note that the Diffusion-QL failed to avoid the risk area as it is not designed for risk-sensitive tasks.

Table 2: Risky robot navigation quantitative evaluation.

| Algorithm | Risky PointMass | | | | Risky Ant | | | |
|---|---|---|---|---|---|---|---|---|
| | Mean | Median | $CVaR_{0.1}$ | Violations | Mean | Median | $CVaR_{0.1}$ | Violations |
| CODAC | -6.1(1.8) | -4.9(1.2) | -14.7(2.4) | 0(0) | -456.3(53.2) | -433.4(47.2) | -686.6(87.2) | 347.8(42.1) |
| ORAAC | -10.7(1.6) | -4.6(1.3) | -64.1(2.6) | 138.7(20.2) | -788.1(98.2) | -795.3(86.2) | -1247.2(105.7) | 1196.1(99.6) |
| CQL | -7.5(2.0) | -4.4(1.6) | -43.4(2.8) | 93.4(10.5) | -967.8(78.2) | -858.5(87.2) | -1887.3(122.4) | 1854.3(130.2) |
| O-WCPG | -12.4(2.5) | -5.1(1.7) | -67.4(4.0) | 123.5(10.6) | -819.2(77.4) | -791.5(80.2) | -1424.5(98.2) | 1455.3(104.5) |
| Diffusion-QL | -13.5(2.2) | -5.7(1.3) | -69.2(4.6) | 140.1(10.5) | -892.5(80.3) | -766.5(66.7) | -1884.4(133.2) | 1902.4(127.9) |
| BEAR | -12.5(1.6) | -5.3(1.2) | -64.3(5.6) | 110.4(9.2) | -823.4(45.2) | -772.5(55.7) | -1424.5(102.3) | 1458.9(100.7) |
| Ours | -5.8(1.4) | -4.2(1.3) | -12.4(2.0) | 0(0) | -392.5(33.2) | -399.5(40.5) | -598.5(50.2) | 291.5(20.8) |

**Performance of UDAC** Our experimental findings indicate that UDAC consistently outperforms other approaches on both the $CVaR_{0.1}$ return and the number of violations, suggesting that UDAC is capable of avoiding risky actions effectively. Furthermore, UDAC exhibits competitive performance in terms of mean return, attributed to its superior performance in $CVaR_{0.1}$. A noteworthy observation is that in the Risky PointMass environment, UDAC learns a safe policy that avoids the risky region entirely, resulting in zero violations, even in the absence of such behavior in the training dataset. In addition, in Section 5.5, we demonstrate that UDAC is effective in optimizing alternative risk-sensitive objectives such as Wang and CPW. Our results illustrate that UDAC can achieve comparable performance to UDAC on $CVaR_{0.1}$ return and the number of violations in the presence of distorted rewards.

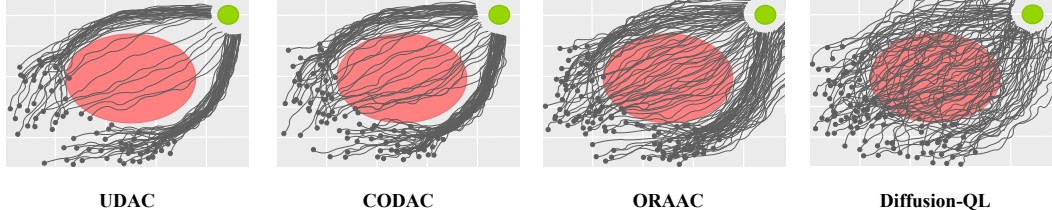

| UDAC | CODAC | ORAAC | Diffusion-QL |

Figure 1: A 2D visualization was created to represent evaluation trajectories on the Risky Ant environment. The risky region is highlighted in red, while the initial states are represented by solid green circles, and the trajectories are shown as grey lines. The results indicate that UDAC consistently approaches the goal while demonstrating the most risk-averse behavior.

**Comparison to ORAAC** ORAAC and UDAC both optimize the CVaR objective, but ORAAC employs imitation learning to constrain the learned policy to the empirical behavior policy. However, this approach may result in suboptimal performance due to the presence of many suboptimal trajectories in the dataset and the risk-neutral nature of the dataset. This issue is particularly relevant in offline RL, where the goal is to leverage large datasets for training that are often heterogeneous in terms of the quality of the behavior policy. In contrast, UDAC is better suited to learn from such datasets by using the diffusion approach, as demonstrated in our results, which show that it outperforms ORAAC in these settings.

**Comparison to CODAC** CODAC employs a strategy that is similar to ORACC by introducing a penalty constant to discourage out-of-distribution actions. Additionally, CODAC adopts a predefined behavior policy to avoid using low-quality behavioral policies in the datasets. However, this approach may lead to suboptimal performance due to the absence of some edge cases. In contrast, UDAC aims to learn to model the behavior policy from the datasets directly via denoising, which has better generalizability. The results suggest that the UDAC can avoid most of the risky areas.

**Comparsion to Diffusion-QL** In environments with significant risks, Diffusion-QL tends to learn policies that exhibit poor tail-performance, which is evident from its low CVaR$_{0.1}$ performance despite the high median performance. Additionally, the mean performance of Diffusion-QL is not satisfactory since it can be heavily influenced by outliers that are not represented in the training dataset. On the other hand, in the Risky Ant environment, Diffusion-QL performs poorly on all metrics but achieves a higher median value than ORAAC. Specifically, Diffusion-QL fails to reach the goal and avoid risky behavior. The majority reason is that the behavior policy used in these environments is risk-neutral and Diffusion-QL shows a strong capability to learn the risk-neutral policy.

## 5.3 RISK-NEUTRAL D4RL

**Tasks** Next, we demonstrate the effectiveness of UDAC even in scenarios where the objective is to optimize the standard expected return. For this purpose, we evaluate UDAC on the widely used D4RL Mujoco benchmark and compare its performance against state-of-the-art algorithms previously benchmarked in Wang et al. (2023). We include ORAAC and CODAC as offline distributional RL baselines. In our evaluation, we report the performance of each algorithm in terms of various metrics, including the mean and median returns, as well as the CVaR$_{0.1}$ returns.

Table 3: Normalized Return on the risk-neutral D4RL Mujoco Suite.

| Dataset | DT | IQL | CQL | Diffusion-QL | ORAAC | CODAC | Ours |
|---|---|---|---|---|---|---|---|
| halfcheetah-medium | 42.6(0.3) | 47.4(0.2) | 44.4(0.5) | 51.1(0.3) | 43.6(0.4) | 48.4(0.4) | 51.7(0.2) |
| hopper-medium | 67.6(6.2) | 66.3(5.7) | 53.0(28.5) | 80.5(4.5) | 10.4(3.5) | 72.1(10.2) | 73.2(8.8) |
| walker2d-medium | 74.0(9.3) | 78.3(8.7) | 73.3(17.7) | 87.0(9.7) | 27.3(6.2) | 82.0(12.3) | 82.7(10.3) |
| halfcheetah-medium-r | 36.6(2.1) | 44.2(1.2) | 45.5(0.7) | 47.8(1.5) | 38.5(2.0) | 48.6(1.8) | 49.0(1.5) |
| hopper-medium-r | 82.7(9.2) | 94.7(8.6) | 88.7(12.9) | 101.3(7.7) | 84.2(10.2) | 95.6(9.7) | 97.2(8.0) |
| walker2d-medium-r | 66.6(4.9) | 73.8(7.1) | 81.8(2.7) | 95.5(5.5) | 69.2(7.2) | 88.2(9.3) | 95.7(6.8) |
| halfcheetah-med-exp | 86.8(4.3) | 86.7(5.3) | 75.6(25.7) | 96.8(5.8) | 24.0(4.3) | 70.4(10.4) | 94.2(8.0) |
| hopper-med-exp | 107.6(11.7) | 91.5(14.3) | 105.6(12.9) | 111.1(12.6) | 28.2(4.5) | 106.0(10.4) | 112.2(10.2) |
| walker2d-med-exp | 108.1(1.9) | 109.6(1.0) | 111.0(1.6) | 110.1(2.0) | 18.2(2.0) | 112.0(6.5) | 114.0(2.3) |

**Results** As per Wang et al. (2023), we directly report results for non-distributional approaches. To evaluate CODAC and ORAAC, we follow the same procedure as Wang et al. (2023), training for 1000 epochs (2000 for Gym tasks) with a batch size of 256 and 1000 gradient steps per epoch. We report results averaged over 5 random seeds. Our experimental results, presented in Table 3, demonstrate that UDAC achieves remarkable performance across all 9 datasets, outperforming the state-of-the-art on 5 datasets (halfcheetah-medium, halfcheetah-medium-replay, walker2d-medium-replay, hopper-medium-expert, and walker2d-medium-expert). Although UDAC performs similarly to Diffusion-QL in risk-neutral tasks, it is designed primarily for risk-sensitive scenarios, and its main strength lies in such situations. In contrast, UDAC leverages diffusion inside the actor to enable expressive behavior modeling and enhance performance compared to CODAC and ORAAC.

## 5.4 HYPERPARAMETER STUDY IN RISK-SENSITIVE D4RL

We intend to investigate a crucial hyperparameter within the actor network, namely $\lambda$, in our ongoing research. In Figure 6, we present an ablation study on the impact of this hyperparameter in risk-sensitive D4RL. Our findings indicate that selecting an appropriate value of $\lambda$ is critical to achieving high performance as it balances pure imitation from the behavior policy with pure reinforcement learning. Specifically, as $\lambda \to 0$, the policy becomes more biased towards imitation and yields poor risk-averse performance, while as $\lambda \to 1$, the policy suffers from bootstrapping errors leading to lower performance. Our experiments suggest that values of $\lambda$ in the range of $[0.05, 0.5]$ tend to perform the best, although the optimal value may vary depending on the specific environment. In all MuJoCo experiments, we set the $\lambda$ parameter, which modulates the action perturbation level, to 0.25 except for the HalfCheetah-medium experiment, where it was set to 0.5. However, as illustrated in Figure 6, our results indicate that these values are not the best for all environments, but instead perform well across most. Due to the page limit, the remaining hyperparameter studies on Risky robot navigation can be found in Appendix D.2.

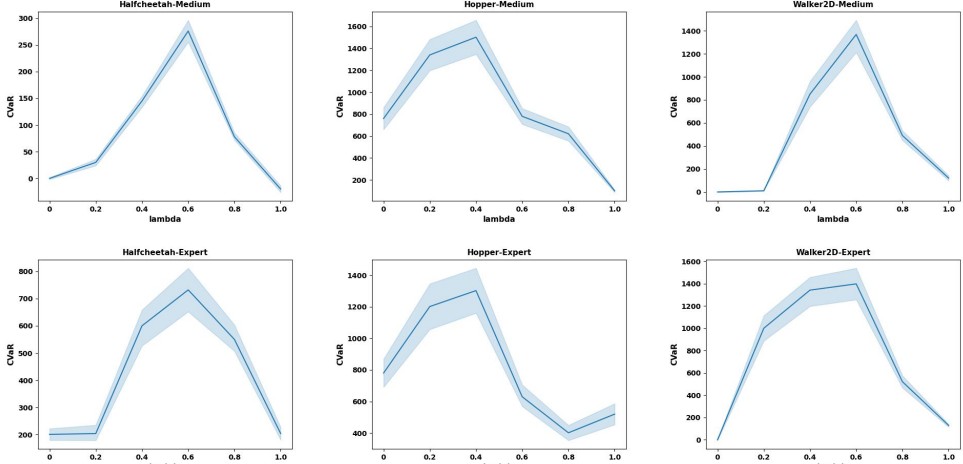

Figure 2: The effect of the hyperparameter $\lambda$ on the CVaR of returns varies across risk-sensitive D4RL environments. When $\lambda$ approaches 0, the policy imitates the behavior policy, resulting in poor risk-averse performance. On the other hand, when $\lambda$ approaches 1, the policy suffers from the bootstrapping error, which leads to low performance. The optimal value of $\lambda$ depends on the environment.

## 5.5 DIFFERENT DISTORTED OPERATORS

In order to demonstrate the performance of UDAC in different distorted operators, we have selected the risky environment - Risky PointMass and Risky Ant to conduct the experiments. We conduct the experiments using a similar setting as mentioned in previous parts. And the results can be found on Table 4. We can find that, the UDAC-Wang achieves similar performance with UDAC in the Risky PointMass but slightly worse on Risky Ant. Based on the conclusions of previous studies (Ma et al., 2021; Dabney et al., 2018a), it can be inferred that Wang's risk preferences are marginally more inclined towards risk-seeking when compared to the CVaR approach. This inference can be attributed to the fact that Wang assigns a non-zero weight (although negligible) to quantile values that exceed the risk cutoff threshold. On the other hand, the CPW approach can be regarded as being similar to a risk-neutral strategy. This is because the CPW approach is designed to model human game-play behavior, and thus does not explicitly incorporate risk preferences into its framework.

Table 4: Various distorted expectation

| Algorithm | Risky PointMass | | | | Risky Ant | | | |
|---|---|---|---|---|---|---|---|---|
| | Mean | Median | $CVaR_{0.1}$ | Violations | Mean | Median | $CVaR_{0.1}$ | Violations |
| UDAC-Wang | -6.1(2.2) | -4.4(1.2) | -15.2(2.4) | 10(3) | -407.5(38.5) | -400.2(41.2) | -653.5(58.2) | 402.5(38.6) |
| UDAC-CPW | -9.4(3.4) | -6.6(2.5) | -56.2(10.6) | 104(22) | -592.5(43.2) | -489.2(47.7) | -726.6(70.4) | 501.6(45.6) |
| UDAC-Neutral | -9.4(3.3) | -4.8(1.5) | -55.2(8.9) | 99(14) | -580.7(55.6) | -458.5(50.5) | -759.6(80.2) | 592.9(57.5) |
| UDAC-CVaR | -5.8(1.4) | -4.2(1.3) | -12.4(2.0) | 0(0) | -392.5(33.2) | -399.5(40.5) | -598.5(50.2) | 291.5(20.8) |

## 6 CONCLUSION AND FUTURE WORK

In this study, we propose a novel model-free offline RL algorithm, namely the Uncertainty-aware offline Distributional Actor-Critic (UDAC). The UDAC framework leverages the diffusion model to enhance its behavior policy modeling capabilities. Our results demonstrate that UDAC outperforms existing risk-averse offline RL methods across various benchmarks, achieving exceptional performance. Moreover, UDAC also achieves comparable performance in risk-natural environments with the recent state-of-the-art methods. Future research directions may include optimizing the sampling speed of the diffusion model (Salimans & Ho, 2022) or other directions related to the optimization of the risk objectives.

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

## A  SOME THEORETICAL ANALYSIS OF THE UDAC

We can observe that the distribution offline RL relies on the Bellman operator $\mathcal{T}^\pi$ and the behavior policy, we have the following important proposition of the UDAC.

**Proposition 1.** *Assume that $Q^\pi$ is the unique fixed point acquired by the $\mathcal{T}^\pi$ as per Theorem 1, then in the support region of the behavior policy $\pi_b$, we denote the Q function of the behavior policy $\pi_b$ as $Q_b$ and the Q function of the learned optimal policy $Q_b^*$. We have $Q_b \le Q^\pi \le Q_b^*$.*

**Proposition 2.** *(Policy Improvement) For all $(s,a) \in \mathcal{S} \times \mathcal{A}$, let $Q^\pi(s,a) = \mathbb{E}(Z^\pi(s,a))$, we have $Q^{\pi_{old}}(s,a) \le Q^{\pi_{new}}(s,a)$ when minimizing the difference between the policy distribution and exponential form of soft action-value function. Where the $\pi$ is defined in Equation (7).*

Proposition 1 guarantees that the learned policy will perform at least as well as behavior policy, and Proposition 2 shows that the soft policy improvement works under the distributional offline RL setting. The proof can be found on the Appendix B.

## B  PROOFS

In order to prove the Proposition 1, we need the following theorem first.

**Theorem 1.** *(Introduced by Bellemare et al. (2017)) The operator $\mathcal{T}^\pi$ is a $\gamma$-contraction in $\overline{d_p}$, which can converge to the optimal action-state value $Q^*$ by taking expectation $\mathbb{E}[Z]$.*

### B.1  PROOF OF THE PROPOSITION 1

*Proof.* According to the Theorem 1, we can find that $Q_b$ is the fixed point of the bellman operator $\mathcal{T}Q(s,a) = R(s,a) + \gamma \mathbb{E}_{s'}\mathbb{E}_{a' \sim \pi_b(\cdot|s')}Q(s',a')$. Similarly, $Q_b^*$ is the fixed point of the optimal bellman operator $\mathcal{T}^*(s,a) = R(s,a) + \gamma \mathbb{E}_{s'}[\max_{a' \sim \pi_b(\cdot|s')} Q(s',a')]$. Then, recall the definition, $\mathcal{T}^\pi Q(s,a) = R(s,a) + \gamma \mathbb{E}_{P(s'|s,a)\pi(a'|s')}[Q(s',a')]$. It's clear that $\mathcal{T}^\pi \ge \mathcal{T}$ and $\mathcal{T}^\pi \le \mathcal{T}^*$. And it leads to $Q_b \le Q^\pi \le Q_b^*$ ☐

### B.2  PROOF OF THE PROPOSITION 2

*Proof.* For any policy $\pi$ (it was defined in Equation (7)) and its distribution $Z^\pi$, we define the $Q^\pi(s,a)$ as,

$$Q^\pi(s,a) = \mathbb{E}(Z^\pi(s,a)) = \mathbb{E}[R(s,a)] + \gamma \mathbb{E}_{P(s'|s,a)\pi(a'|s')}[Q(s',a')]. \tag{12}$$

Now we have the old policy $\pi_{old}$ and its Q value $Q^{\pi_{old}}$. We obtain the new policy $\pi_{new}$ by minimizing the following difference (i.e.,the KL-divergence),

$$\pi_{new}(\cdot|s) = \arg\min_{\pi'} D_{KL}\Big(\pi'(\cdot|s)\|\frac{\exp\left(Q^{\pi_{old}}(s,\cdot)\right)}{\Delta^{old}(s))}\Big). \tag{13}$$

The $\Delta^{old}$ is used to normalize the distribution. It can be transformed into the following,

$$\pi_{new}(\cdot|s) = \arg\min_{\pi'} D_{KL}\Big(\pi'(\cdot|s)\|\exp\left(Q^{\pi_{old}}(s,\cdot)\right) - \log\Delta^{old}(s))\Big). \tag{14}$$

Since, $\pi^{new}$ is the solution of the above equation, we have,

$$\mathbb{E}_{a\sim\pi_{new}(\cdot|s)}\Big[\log\pi_{new}(a|s) - Q^{\pi_{old}}(s,a) + \log\Delta^{old}(s)\Big] \tag{15}$$

$$\le \mathbb{E}_{a\sim\pi_{old}(\cdot|s)}\Big[\log\pi_{old}(a|s) - Q^{\pi_{old}}(s,a) + \log\Delta^{old}(s)\Big]. \tag{16}$$

And we can further remove the $\Delta^{old}(s)$ as it only related to $s$,

$$\mathbb{E}_{a\sim\pi_{new}(\cdot|s)}\Big[\log\pi_{new}(a|s) - Q^{\pi_{old}}(s,a)\Big] \le \mathbb{E}_{a\sim\pi_{old}(\cdot|s)}\Big[\log\pi_{old}(a|s) - Q^{\pi_{old}}(s,a)\Big]. \tag{17}$$

Then, we extend it into the Bellman equation,

$$Q^{\pi_{old}}(s_t, a_t) = \mathbb{E}(R(s_t, a_t)) + \gamma \mathbb{E}_{s_{t+1} \sim P(\cdot|s_t, a_t), a_{t+1} \sim \pi_{old}(\cdot|s_{t+1})}[Q^{\pi_{old}}(s_{t+1}, a_{t+1})] \quad (18)$$

$$\leq \mathbb{E}(R(s_t, a_t)) + \gamma \mathbb{E}_{s_{t+1} \sim P(\cdot|s_t, a_t), a_{t+1} \sim \pi_{new}(\cdot|s_{t+1})}[Q^{\pi_{old}}(s_{t+1}, a_{t+1})] \quad (19)$$

$$\vdots \quad (20)$$

$$\leq Q^{\pi_{new}}(s_t, a_t) \quad (21)$$

$$(22)$$

$$\square$$

## C  IMPLEMENTATION DETAILS

In this section, we will describe the implementation details of the proposed UDAC.

**Actor**  As mentioned in the main text, the architecture of the actor is,

$$\pi_\phi(s) = \lambda \xi(\cdot|s, \beta) + \beta, \beta \sim \pi_b,$$

where $\xi : \mathcal{A} \to \mathbb{R}^{\|\mathcal{A}\|}$ (i.e., the traditional Q-learning component). And $\beta$ is the output of the behavioral model (i.e., the diffusion component).

If we want to integrate the proposed UDAC into an online setting, we can effortlessly remove the $\beta$ and set $\lambda = 1$. For our experiments, we construct the behavioral policy as an MLP-based conditional diffusion model, drawing inspiration from Ho et al. (2020); Nichol & Dhariwal (2021). We employ a 3-layer MLP with Mish activations and 256 hidden units to model $\epsilon_\psi$. The input of $\epsilon_\psi$ is a concatenation of the last step's action, the current state vector, and the sinusoidal positional embedding of timestep $i$. The output of $\epsilon_\psi$ is the predicted residual at diffusion timestep $i$. Moreover, we build two Q-networks with the same MLP setting as our diffusion policy, with 3-layer MLPs with Mish activations and 256 hidden units for all networks. Moreover, as mentioned in the main body, we set $N$ to be small to relieve the bottleneck caused by the diffusion. In practice, we use the noise schedule obtained under the variance preserving SDE (Song et al., 2020).

**Critic**  For the critic architecture, we just follow the instruction on the ORAAC (Urpí et al., 2021) with continuous action.

We use Adam optimizer to optimize the proposed model. The learning rate is set to 0.001 for the critic and the diffusion, 0.0001 for the actor model. The target networks for the critic and the perturbation models are updated softly with $\mu = 0.005$.

## D  EXPERIMENTS DETAILS AND EXTRA RESULTS

### D.1  EXPERIMENTS SETUP

The stochastic MuJoCo environments are set as follows with a stochastic reward modification. It was proposed by Urpí et al. (2021), and the following description is adapted:

- Half-Cheetah: $R_t(s, a) = \overline{r}_t(s, a) - 70\mathcal{I}_{v > \overline{v}} \cdot \mathcal{B}_{0.1}$. where $\overline{r}_t(s, a)$ is the original environment reward, $v$ is the forward velocity, and $\overline{v}$ is a threshold velocity ($\overline{v} = 4$ for Medium datasets and $\overline{v} = 10$ for the Expert dataset). The maximum episode length is reduced to 200 steps.

- Walker2D/Hopper: $R_t(s, a) = \overline{r}_t(s, a) - p\mathcal{I}_{|\theta| > \overline{\theta}} \cdot \mathcal{B}_{0.1}$, where $\overline{r}_t(s, a)$ is the original environment reward, $\theta$ is the pitch angle, and $\overline{\theta}$ is a threshold velocity ($\overline{\theta} = 0.5$ for Walker2D and $\overline{\theta} = 0.1$ Hopper). And $p = 30$ for Walker2D and $p = 50$ for Hopper. When $|\theta| > 2\overline{\theta}$ the robot falls, the episode terminates. The maximum episode length is reduced to 500 steps.

The risky robot navigation is proposed by Ma et al. (2021), we just provide an illustration here (see Figure 3) to better understand the goal of this task.

The experiments are conducted on a server with two Intel Xeon CPU E5-2697 v2 CPUs with 6 NVIDIA TITAN X Pascal GPUs, 2 NVIDIA TITAN RTX, and 768 GB memory.

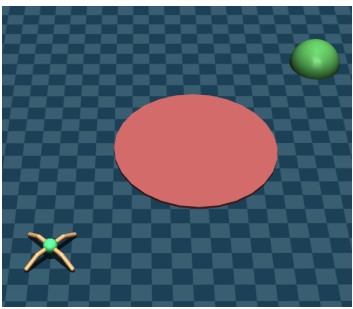

Figure 3: Risky robot navigation

## D.2 HYPERPARAMETERS

As the UDAC builds on top of ORAAC, we just keep ORAAC-specific hyperparameters identical to the original work. For the remaining hyperparameters such as the learning rate and whether to use max Q backup, please refers to our source code which is provided. Moreover, here we also provide the hyperparameters study on the risky robot tasks.

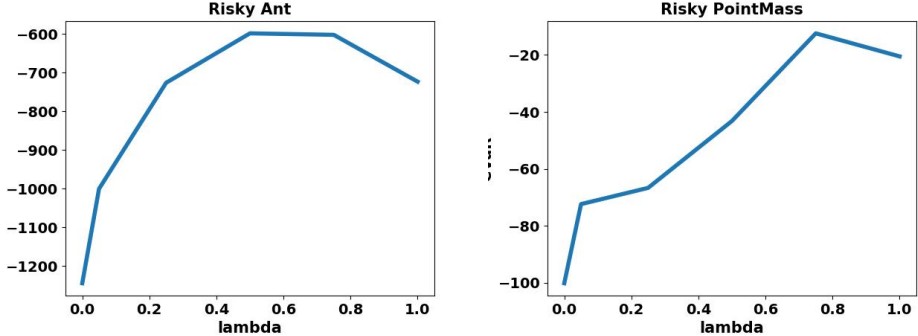

Figure 4: The effect of the hyperparameter $\lambda$ on the CVaR of returns varies across risky robot navigation environments.

## D.3 EXTRA EXPERIMENTS

We also conduct the experiments on the mixed datasets.

Table 5: Performance of the UDAC algorithm on risk-sensitive offline D4RL environments using medium, expert and mixed datasets. We compare the $CVaR_{0.1}$ and mean of the episode returns, and report the mean (standard deviation) of each metric. Our results indicate that across all environments, UDAC outperforms benchmarks with respect to $CVaR_{0.1}$. Furthermore, in environments that terminate, UDAC achieves longer episode durations as well.

| | Algorithm | Medium | | Expert | | Mixed | |
|---|---|---|---|---|---|---|---|
| | | $CVaR_{0.1}$ | Mean | $CVaR_{0.1}$ | Mean | $CVaR_{0.1}$ | Mean |
| Half-Cheetah | O-RAAC | 214(36)* | 331(30) | 595(191) | 1180(78) | 307(6) | 119(27) |
| | CODAC | -41(17) | 338(25) | 687(152)* | 1255(101)* | 396(56)* | 238(59)* |
| | O-WCPG | 76(14) | 316(23) | 248(232) | 905(107) | 217(33) | 164(76) |
| | Diffusion-QL | 84(32) | 329(20)* | 542(102) | 728(104) | 199(43) | 143(77) |
| | BEAR | 15(30) | 312(20) | 44(20) | 557(15) | 24(30) | 324(59) |
| | CQL | -15(17) | 33(36) | -207(47) | -75(23) | 214(52) | 12(24) |
| | Ours | **276(29)** | **417(18)** | **732(104)** | **1352(100)** | **462(48)** | **275(50)** |
| Walker-2D | O-RAAC | 663(124) | 1134(20) | 1172(71) | 2006(56) | 222(37) | -70(76) |
| | CODAC | 1159(57) | 1537(78)* | 1298(98)* | 2102(102)* | 450(193)* | 261(231)* |
| | O-WCPG | -15(41) | 283(37) | 362(33) | 1372(160) | 201(33) | -77(54) |
| | Diffusion-QL | 31(10) | 273(20) | 621(49) | 1202(63) | 302(154) | 102(88) |
| | BEAR | 517(66) | 1318(31) | 1017(49) | 1783(32) | 291(90) | 58(89) |
| | CQL | 1244(128)* | 1524(99) | 1301(78) | 2018(65) | 74(77) | -64(-78) |
| | Ours | **1368(89)** | **1602(85)** | **1398(67)** | **2198(78)** | **498(91)** | **299(194)** |
| Hopper | O-RAAC | 1416(28)* | 1482(4) | 980(28) | 1385(33) | 876(87) | 525(323) |
| | CODAC | 976(30) | 1014(16) | 990(19) | 1398(29) | 1551(33)* | 1450(101)* |
| | O-WCPG | -87(25) | 69(8) | 720(34) | 898(12) | 372(100) | 442(201) |
| | Diffusion-QL | 981(28) | 1001(11) | 406(31) | 583(18) | 1021(102) | 1010(59) |
| | BEAR | 1252(47) | 1575(8)* | 852(30) | 1180(12) | 758(128) | 462(202) |
| | CQL | 1344(40) | 1524(20) | 1289(30)* | 1402(30)* | 189(63) | -21(62) |
| | Ours | **1502(19)** | **1602(18)** | **1302(28)** | **1502(27)** | **1620(42)** | **1523(90)** |

