# OpenReview forum: "Uncertainty-aware Distributional Offline Reinforcement Learning"
_ICLR.cc/2024/Conference — Submitted to ICLR 2024_

### Official Review · Reviewer_PL45 · 2023-10-31

**Soundness:** 3 good
**Presentation:** 3 good
**Contribution:** 2 fair
**Rating:** 6
**Confidence:** 4

**Summary:**

This paper proposes to mitigate both epistemic and aleatoric uncertainty in offline reinforcement learning by combining risk-sensitive RL and diffusion models. Specifically, the authors used a distributional critic to estimate risk measures and a diffusion model to model behavior policy.

**Strengths:**

1. Although some theoretical results and methods are mainly from prior works, this paper is still well-motivated and proposes a good method.
2. The experiments generally do a good job of illustrating the more risk-sensitive performance of the algorithm. Results show that UDAC performs well on risk-sensitive and risk-neutral tasks.

**Weaknesses:**

Major issues:
1. The paper uses a diffusion model, for the latter performs better than VAE, under the condition of a dataset with different modalities(in Section 4.2). This part seems to lack theoretical analysis for ”denoising a noisy mixed policy. How does the diffusion model decide which part of the policy is noisy? Or it's just the diffusion model has a better representational ability? Some theoretical analysis or a numerical one would be better.
2. In eq.(7) the policy $\pi_\phi$ does not take the risk distortion parameter as an input, does this imply that the policy's risk level can't be tuned? If it can be tuned, studying this would be more innovative than studying $\lambda$.
Minor issues:
1. In last paragraph of Section 1, DR4L should be D4RL.

**Questions:**

1. Please clarify how diffusion model learns from the behavior policy $\pi_b$ instead of cloning it.
2. What are $\alpha_i, \overline{\alpha}_i$ in eq.(9) (10) (11)?
3. Does the policy learn a fixed risk-averse distortion? If so, can the network be modifed to accept different risk level in test time?

---

> ### Author Response · Authors · 2023-11-15
> **Response to Reviewer PL45 - Part 1**
>
> We sincerely appreciate the reviewers' thorough evaluation of our paper. We acknowledge the concerns that have been raised regarding clarity, technical details, and experimental aspects. We have conducted a thorough revision to address these concerns. We have provided a detailed point-by-point response to your comments.
>
> >The paper uses a diffusion model, for the latter performs better than VAE, under the condition of a dataset with different modalities(in Section 4.2). This part seems to lack theoretical analysis for ”denoising a noisy mixed policy. How does the diffusion model decide which part of the policy is noisy? Or it's just the diffusion model has a better representational ability? Some theoretical analysis or a numerical one would be better.
>
> We appreciate the reviewer's concerns. Our model operates under the assumption that the pre-collected trajectories form a mixture of different behavior policies, some of which may be sub-optimal.
> Regrettably, from a denoising perspective, to our knowledge in the RL domain, we are not aware of a close-up of existing works that try to control the segment that needs to be denoised. We believe this a valuable point, we will explore the potential relevant works in other domains such as CV to investigate the feasibility.
> Given that the diffusion model is trained together with the RL agent, we think the diffusion process can identify noisy segments based on the rewards obtained during ongoing training. We believe that diffusion offers superior representational capabilities compared to methods like VAE or cVAE within our target domain and our experiments also substantiate this claim. For instance, in a baseline comparison, ORAAC, utilizing VAE for behavior policy representation, displays lower performance across all tasks compared to our proposed method. Additionally, recent studies (Pearce et al., 2023; Wang et al., 2023) corroborate the assertion that diffusion possesses stronger representational abilities than VAEs. Regarding the numerical experiments, we are uncertain about the specific expectations of the reviewers. It would greatly assist us if the reviewers could provide more detailed specifications. Thanks again for your thoughtful review, we are committed to advancing the point that control the diffusion model to denoise certain segments in the RL domain.
>
> Tim Pearce, Tabish Rashid, Anssi Kanervisto, Dave Bignell, Mingfei Sun, Raluca Georgescu,
> Sergio Valcarcel Macua, Shan Zheng Tan, Ida Momennejad, Katja Hofmann, and Sam De-
> vlin. Imitating human behaviour with diffusion models. ICLR, 2023
>
> Zhendong Wang, Jonathan J Hunt, and Mingyuan Zhou. Diffusion policies as an expressive policy class for offline reinforcement learning. ICLR, 2023
>
> > In eq.(7) the policy does not take the risk distortion parameter as an input, does this imply that the policy's risk level can't be tuned? If it can be tuned, studying this would be more innovative than studying $\lambda$
>
> We appreciate the reviewer's concern. The purpose of eq.(7) is used to show how the actor's policy $\pi_\phi(s)$ is composed. The risk distortion parameter is an input and will be used to train the $\pi_\phi(s)$ via eq.(6). We hope this justification can address your concern.
>
> > Minor issues: In the last paragraph of Section 1, DR4L should be D4RL.
>
> We appreciate the reviewer's comment. We have revised the typos and checked the whole paper to ensure all the typos are revised.
>
> >Please clarify how diffusion model learns from the behavior policy  instead of cloning it.
>
> We appreciate the reviewer's concern. As the diffusion process is kind of a denoising process, it indeed will perform like behavior cloning in the context of RL. However, it deviates from traditional behavior cloning in its approach. Instead of replicating the demonstrations directly, the diffusion method identifies segments of the data as noise and excludes them. This denoising process results in a learned representation that differs from the initial pre-collected data, having refined segments. This process leans more toward imitation rather than strict behavior cloning.
>
> > What are  $\alpha_i, \overline{\alpha}_i$in eq.(9) (10) (11)?
>
>  We appreciate the reviewer's comment. The variables $\alpha_i$ and $\overline{\alpha}_i$are directly linked to the forward process variances $\beta_i$. Specifically, $\alpha_i = 1-\beta_i$,  $\overline{\alpha}_i$ is defined as the sum of $\alpha_i$ from 0 to $t$ . We chose to omit this explanation as it mirrors the conventional diffusion model. Our intention was to conserve space by refraining from detailing how the MDP integrates into the diffusion process.

---

> ### Author Response · Authors · 2023-11-15
> **Response to Reviewer PL45 - Part 2**
>
> > Does the policy learn a fixed risk-averse distortion? If so, can the network be modifed to accept different risk level in test time?
>
> We appreciate the reviewer's concern. In the current version, the proposed method needs to be optimized based on certain risk distortions (such as $CVaR_\alpha$, Wang etc). Our model can be modified to accept different $\alpha$ in CVaR, given that the $\alpha$ is a hyperparameter of the proposed method. Specifically, we can change the $\alpha$ of CVaR in Equation 6 (if we are using CVaR as the distortion). For your reference, the $CVaR_\alpha$ is defined as,
>
> $$
>  CVaR_\alpha (F_Z^{-1}(s,a;\tau)) = \frac{1}{\alpha} \int^{\alpha}_0 F_Z^{-1}(s,a;\tau)d \tau
> $$
>
> where $a$ is the output of the deterministic policy $\pi$ with parameter $\phi$.
> Moreover, it would also be possible to adjust the risk level by modifying the testing environments. For example, we can adjust the stochastic reward setting in risk-sensitive D4RL to make it more risky.However, considering the alterations made to the test environments, it's possible that the effectiveness of the proposed method might be impacted. We view this as a potential avenue for promising future research. For your reference, we have also provided some experimental results about different $\alpha$ of CVaR.
>
> |              |           |                     |                    |                     |                     |                    |                     |
> |:------------:|:---------:|:-------------------:|:------------------:|:-------------------:|:-------------------:|:------------------:|:-------------------:|
> |              | Algorithm |       Medium        |                    |                     |       Expert        |                    |                     |
> |              |           | $CVaR_{0.05}$ | $CVaR_{0.1}$  | $CVaR_{0.15}$  | $CVaR_{0.05}$  | $CVaR_{0.1}$  | $CVaR_{0.15}$  |
> | Half-Cheetah |   UDAC    |       200(14)       |      276(29)       |       303(35)       |       602(55)       |      732(104)      |      782(201)       |
> |              |   ORAAC   |       188(30)       |      214(36)       |       277(43)       |      588(104)       |      595(191)      |       599(99)       |
> |              |   CODAC   |       -33(12)       |      -41(17)       |       -40(10)       |      620(104)       |      687(152)      |       699(83)       |
> |  Walker-2D   |   UDAC    |      1242(100)      |      1368(89)      |      1343(165)      |      1257(94)       |      1398(67)      |      1402(178)      |
> |              |   ORAAC   |      610(130)       |      663(124)      |       732(99)       |      1086(59)       |      1172(71)      |      1153(91)       |
> |              |   CODAC   |      1002(89)       |      1159(57)      |      1172(100)      |      1190(73)       |      1298(98)      |      1320(64)       |
> |    Hopper    |   UDAC    |      1443(41)       |      1502(19)      |      1742(44)       |      1267(50)       |      1302(28)      |      1399(34)       |
> |              |   ORAAC   |      1388(34)       |      1416(28)      |      1423(44)       |       853(40)       |      980(28)       |       992(30)       |
> |              |   CODAC   |       955(28)       |      976(30)       |       988(25)       |       903(44)       |      990(19)       |      1001(32)       |

---

### Official Review · Reviewer_sAmU · 2023-11-05

**Soundness:** 3 good
**Presentation:** 2 fair
**Contribution:** 2 fair
**Rating:** 6
**Confidence:** 3

**Summary:**

This paper proposes a model-free RL algorithm for learning risk-averse policies in offline RL setting. This algorithm uses distributional RL framework for obtaining risk-aversion to aleatoric uncertainty and diffusion model for modelling the behavior policy.
The authors claim that the diffusion model is better able to capture the different sub-optimal behavior policies used to generate the data than VAEs.
The authors show empirically that the proposed algorithm outperforms many of the existing offline RL algorithms in many risk-sensitive D4RL and risky robot navigation task and has comparable performance on many risk-neutral D4RL environments.

**Strengths:**

The authors evaluate the proposed algorithm on several risk-sensitive D4RL environments , Robot Navigation Task, risk-neutral D4RL environments and compares them with several baselines from recent work.

The authors also provide ablation studies on the effect of using different distortion operators and hyperparameters on the performance of the proposed framework.

**Weaknesses:**

The paper is not clearly written, for example, see first 4 lines of section 4.1.

Novelty is limited since risk-averse distributional offline RL have been explored by other work like [1].  This work only extends prior work by using diffusion model to model the behavior policy instead of VAEs.

The authors claim that using their framework accurately models the behavior policy but does not provide any theoretical justification for the claim.

The proposed algorithm and baselines are compared at a single CVaR confidence level $\alpha=0.1$. It would be good to evaluate them on different confidence levels.

While reporting the performance of the proposed algorithm and baselines, the authors do not provide the errors in the mean and cvar value of returns.


[1] https://arxiv.org/pdf/2102.05371.pdf

Additional Comments:
The first line of 4.1 is grammatically incorrect.

noise policies (section 4.2)  -> noisy policies

**Questions:**

How does this framework guarantee robustness against epistemic uncertainty without explicitly optimizing any robust measure of expected returns?

Why choose confidence level $\alpha=0.1$ while evaluating the performance of the framework with CVaR metric?

---

> ### Author Response · Authors · 2023-11-15
> **Response to Reviewer sAmU - Part 1**
>
> We sincerely appreciate the reviewers' thorough evaluation of our paper. We acknowledge the concerns that have been raised regarding clarity, technical details, and experimental aspects. We have conducted a thorough revision to address these concerns. We have provided a detailed point-by-point response to your comments.
>
> >  the paper is not clearly written, for example, see first 4 lines of section 4.1.
>
> We genuinely appreciate the reviewer's concerns. In order to improve the clarity, we have revised the first paragraph of section 4.1.
>
> >Novelty is limited since risk-averse distributional offline RL has been explored by other work like [1]. This work only extends prior work by using diffusion model to model the behavior policy instead of VAEs.
>
> We appreciate the reviewer’s concern. We would like to make a justification for the novelty. As the reviewer pointed out, our work is an extension of the existing work [1] (i.e., ORAAC). The biggest limitation of the ORAAC would be the expressive capability of the behavior policy model. Moreover, replacing the VAE with the diffusion model is not an easy task. The diffusion model we used here is not just a standard DDPM. We define the overall diffusion as a conditional model with states as the condition and actions as the outputs. Different from the ORAAC which trains the VAE separately from the RL agent, UDAC trains the diffusion model and the RL agent simultaneously. By using such a way, UDAC is able to understand and adapt to the environment dynamic (i.e., aleatoric uncertainty) efficiently.  Different from vanilla DDPM, the diffusion model used in UDAC only requires a few steps (less than 3) to learn a superior behavior policy model.
>
> > The authors claim that using their framework accurately models the behavior policy but does not provide any theoretical justification for the claim.
>
> We appreciate reviewer’s concern regarding the theoretical justification for the accurate behavior policy modeling.
> In this study, we mainly focus on validating the quality of the learned behavior representation through comprehensive experiments. We can find that the proposed method outperforms than existing baselines in both risk-sensitive environments and risky environments.
> To the best of our knowledge, it is challenging to theoretically prove that the proposed behavior policy model is more accurate than the existing works, since the recent works (Pearce et al., 2023; Wang et al., 2023) about the theoretical justification of diffusion in RL for representation learning are still unexplored. In this work, we have provided some theoretical analysis of the UDAC from the convergence perspective and policy improvement perspective (please kindly refer to, Appendix A, Proposition 1 and Proposition 2). Given our empirical results along with theoretical analysis, we can conclude that the proposed behavior policy modeling method can learn the behavior policy more accurately or expressively.
>
> Tim Pearce, Tabish Rashid, Anssi Kanervisto, Dave Bignell, Mingfei Sun, Raluca Georgescu, Sergio Valcarcel Macua, Shan Zheng Tan, Ida Momennejad, Katja Hofmann, and Sam Devlin. Imitating human behaviour with diffusion models. ICLR, 2023
>
> Zhendong Wang, Jonathan J Hunt, and Mingyuan Zhou. Diffusion policies as an expressive policy class for offline reinforcement learning. ICLR, 2023

---

> ### Author Response · Authors · 2023-11-15
> **Response to Reviewer sAmU - Part 2**
>
> >  The proposed algorithm and baselines are compared at a single CVaR confidence level $\alpha=0.1$. It would be good to evaluate them on different confidence levels.
>
> We appreciate the reviewer’s comment. Given that all the existing baselines are evaluated under the single CvaR confidence level 0.1, we choose to use 0.1 solely. To address your concern, we have run the different levels of CvaR on the proposed model. We choose the alpha from the range 0.05, 0.1, and 0.15 which represent the CvaR at the 5\%, 10\% and 15\% quantile levels. Given the timely constraint of the rebuttal phase, we are not able to provide a comprehensive comparison of all of the baselines under different quantile levels. As a result, we have reported the most two relevant baselines for your reference (i.e., ORAAC and CODAC). Results are attached to the second part of the response.
>
> |              |           |                     |                    |                     |                     |                    |                     |
> |:------------:|:---------:|:-------------------:|:------------------:|:-------------------:|:-------------------:|:------------------:|:-------------------:|
> |              | Algorithm |       Medium        |                    |                     |       Expert        |                    |                     |
> |              |           | $CVaR_0.05$ | $CVaR_0.1$ |$CVaR_0.15$ | $CVaR_0.05$ |$CVaR_0.1$| $CVaR_0.15$ |
> | Half-Cheetah |   UDAC    |       200(14)       |      276(29)       |       303(35)       |       602(55)       |      732(104)      |      782(201)       |
> |              |   ORAAC   |       188(30)       |      214(36)       |       277(43)       |      588(104)       |      595(191)      |       599(99)       |
> |              |   CODAC   |       -33(12)       |      -41(17)       |       -40(10)       |      620(104)       |      687(152)      |       699(83)       |
> |  Walker-2D   |   UDAC    |      1242(100)      |      1368(89)      |      1343(165)      |      1257(94)       |      1398(67)      |      1402(178)      |
> |              |   ORAAC   |      610(130)       |      663(124)      |       732(99)       |      1086(59)       |      1172(71)      |      1153(91)       |
> |              |   CODAC   |      1002(89)       |      1159(57)      |      1172(100)      |      1190(73)       |      1298(98)      |      1320(64)       |
> |    Hopper    |   UDAC    |      1443(41)       |      1502(19)      |      1742(44)       |      1267(50)       |      1302(28)      |      1399(34)       |
> |              |   ORAAC   |      1388(34)       |      1416(28)      |      1423(44)       |       853(40)       |      980(28)       |       992(30)       |
> |              |   CODAC   |       955(28)       |      976(30)       |       988(25)       |       903(44)       |      990(19)       |      1001(32)       |
>
> >  While reporting the performance of the proposed algorithm and baselines, the authors do not provide the errors in the mean and cvar value of returns.
>
>  We appreciate the reviewer’s comment. We would appreciate it if the reviewer could give more details about this concern. As we reported in Table 1 and Table 2, all the reported values are in the form `mean(error)’. Moreover, we have added the errors in Table 4 in the revised version. We hope this can address your concern.
>
> > Additional Comments: The first line of 4.1 is grammatically incorrect.
>
> We appreciate the reviewer’s comment. In order to improve the clarity, we have revised the first paragraph of section 4.1
>
> >noise policies (section 4.2) $\rightarrow$ noisy policies
>
>  We appreciate the reviewer’s comment. We have revised this typo.

---

> ### Comment · Reviewer_sAmU · 2023-11-16
>
> Thank you for the clarification and for responding to my concerns.
> The theory in the appendix is not specific to the behavior policy modelling and it just shows that the distributional bellman operator is a contraction mapping and proves that the distributional RL algorithm will improve the policy over time.
> I am not very convinced that simple replacing one generative model (VAE) with another (Diffusion model) is a novel enough contribution for this conference.
>
> Can you please also answer the following question:
>
> a) How does this framework guarantee robustness against epistemic uncertainty without explicitly optimizing any robust measure of expected returns?

---

> ### Author Response · Authors · 2023-11-17
> **Response to Reviewe sAmU**
>
> We are pleased to know that our response has successfully addressed the  majority of your concerns. We agree that the provided theoretical analysis may not  specifically related to the behavior modeling. Given that the behavior modeling is kind of a representation learning, we decide to use the experiments to valid the feasibility.  Considering the breadth of comprehensive experiments we have conducted, which demonstrate the superior performance of our proposed method over baseline models, we believe that our work can offer some valuable insights to the community.
>
> Regarding your additional concern, please refers to the following response.
>
> > How does this framework guarantee robustness against epistemic uncertainty without explicitly optimizing any robust measure of expected returns?
>
> We appreciate your questions. We acknowledge the epistemic uncertainty arising from limited data[1]. When constructing a model with this dataset for future state or reward prediction, there is a risk of bias if we solely rely on immediate returns as our optimization objective. In this work, we seek distributional reinforcement learning, which optimizes the distribution of states and rewards rather than individual outcomes. This approach enhances the model's ability to learn about unseen states, effectively mitigating the impact of epistemic uncertainty, and thereby ensuring a certain degree of robustness.
>
> [1] Clements, W. R., Van Delft, B., Robaglia, B. M., Slaoui, R. B., & Toth, S. (2019). Estimating risk and uncertainty in deep reinforcement learning. arXiv preprint arXiv:1905.09638.

---

> > ### Comment · Reviewer_sAmU · 2023-11-21
> >
> > Thank you for your comments. I think the uncertainty that is being addressed is aleatoric uncertainty and not epistemic.

---

> > > ### Author Response · Authors · 2023-11-21
> > >
> > > Thank you for your advice about the aleatoric uncertainty and epistemic uncertainty. We would like to sincerely apologize for the confusion between the aleatoric uncertainty and epistemic uncertainty. In the proposed method, we are targeting epistemic uncertainty from the risk-sensitive perspective (i.e., risk aversion). The proposed method utilizes the perturbation model to ask the network to learn a risk-sensitive policy that can reduce the epistemic uncertainty.

---

### Official Review · Reviewer_LtLP · 2023-11-06

**Soundness:** 2 fair
**Presentation:** 2 fair
**Contribution:** 2 fair
**Rating:** 3
**Confidence:** 4

**Summary:**

The paper introduces UDAC (Uncertainty Aware offline Distributional Actor Critic), a distrubutional algorithm that tackles the problem of risk-averse policy learning via introduction of a risk-distortion function in the actor loss.  A typical probem in offline RL is that algorithms suffer from OOD issues, an attempt to remedy this is to stay close to the data distribution. This paper achieves this by decomposing the actor into a behavior policy part and perturbation part, the behavior policy is learned by a diffusion generative model and perturbation part is learnable by the distributional risk-averse RL loss. The contribution of the policy is controlled by a hyperparameter $\lambda$. The method achieves SotA results in risk-averseness on multiple benchmarks and is comparable to risk-netural algorithms on D4RL, some sensitivity analysis is provided.

**Strengths:**

Fairly extensive experimental evaluation and results that beat other methods.

Perturbation formulation for trading-off staying on data vs. doing pure RL seems novel to me, but I could be wrong here since it is a very simple modification.

The method at its core is simple to understand.

**Weaknesses:**

There are a lot of errors in the math that is written in the paper (see questions).

Typos are also very prevalent, I didn't enumerate all of them.

No variance of the runs is reported so I cannot judge how stable the method is.

There are a lot of technical things in the paper that are not necessary and prevent the space from being used for better analysis of the method (there are some propositions in the appendix for example that don't appear in the main text). On the other hand, there are some terms used that are not defined.

The novelty of the paper is very limited (however, since the results are signficant I don't consider this an absolutely major flaw)

Justification for score: I think that the way the paper is written doesn't meet the bar for conference publication (with the equation flaws, typos and technical details that don't focus on the main contributions).

**Questions:**

I think that i) Elimination of Manual Behavior Policy Specification. ii) Accurate Modeling of the behavior policy is the same statement? You use diffusion generative models to model the behavior policy, that's it?

p2.par2 - "We conducted extensive experiments in three different environments covering both risk-sensitive" - it's 3 different benchmarks, you use multiple environments of D4RL.

p3. par2 - "For simplify" - to simplify

In Section 4 and earlier you talk about "risk-distortion operators", but you do not define the concept in the background section, yet it seems essential for your method.

In Section 4.1 you talk about implicit quantile functions, yet this concept has not been defined (what is implicit about them?) and no reference is given, I suppose you need [1].

There is no citation for the quantile Huber loss, you should provide a citation for it and refer to the equations the first time it is mentioned (ideally the definition would preceed  the first usage of the function).

p3, last par - you say that you use both uniform sampling and a quantile proposal network, then you say that you don't use a quantile proposal network and leave it for future work? If you don't use it, don't mention it here or mention it as a comment (an alternative is to use a quantile proposal network such as in (xyz citation)

In eq. 5, I suppose that you have N fixed quantile levels $\tau_i$, hence the indexing. However, this was not mentioned previously, yet it is part of the methodology (also along the lines of earlier work?). Moreover, the loss is a function o $\theta$ and there is no expectation defined over s, a, s', a', though the quantile td error depends on the former. Also, "J" is commonly used as return? Also normalizing factor should be $N^2$?

eq. 6 - is it "distortion risk function" or "risk-distortion function"? I also don't understand this loss function, you have expectations wrt. marginal state distribution, however you have an action appearing in the implicit quantile function, where does the action come from? You are missing a policy here, please define this correctly.

I think that your description of diffusion generative modelling needs significant adjustments. What you call an $\epsilon$-model is the denoiser, the Gaussian reverse process transition kernel arises from a continuous treatment of diffusion and exactly because of the choice of noise schedule. Since you don't provide contributions on level of diffusion generative modelling, this shouldn't take so much space of the paper.

p6 -   CQL is not a distributional RL algorithm?

suggestion for table 1 - use colors instead of bold and *

figure 2 needs correction, y axis is not visible, also there are no error bands - how do we know what is the stability across reruns?

Is the diffusion generative model absolutely necessary in order to achieve this performance? Would you switch it out for a CVAE in order to compare?

[1] Will Dabney, Georg Ostrovski, David Silver, Rémi Munos, "Implicit Quantile Networks for Distributional Reinforcement Learning"

---

> ### Author Response · Authors · 2023-11-15
> **Response to Reviewer LtLP - Part 1**
>
> We sincerely appreciate the reviewers' thorough evaluation of our paper.  We acknowledge the concerns that have been raised regarding clarity, citations, and experimental aspects. We have conducted a thorough revision to address these concerns. We have provided a detailed point-by-point response to your comments.
>
> > Elimination of Manual Behavior Policy Specification. ii) Accurate Modeling of the behavior policy is the same statement? You use diffusion generative models to model the behavior policy, that's it?}
>
> We thank for reviewer’s comment. We would like to provide a further clarification about these two statements. The first one refers to the previous work (Ma et al. 2021) which requires us to define the behavior policy manually. To achieve this, we can use cVAE or diffusion model (as what we do in this work) to avoid the manual definition. The second statement refers to the previous works (Urpi et al., 2021 and Lyu et al., 2022) which employ the VAE/cVAE to represent the behavior policy.  The reason why we do not omit the first statement is because the performance of the Ma et al. 2021 is quite good in some tasks (we would kindly ask the reviewer to refer the table 1). Hence, we decided to keep this statement as one of the main streams in the current progress of the distributional RL.
> Moreover, replacing the VAE with the diffusion model is not an easy task. The diffusion model we used here is not just a standard DDPM. We define the overall diffusion as a conditional model with states as the condition and actions as the outputs. Different from the ORAAC which trains the VAE separately from the RL agent, UDAC trains the diffusion model and the RL agent simultaneously. By using such a way, UDAC is able to understand and adapt to the environment dynamic (i.e., aleatoric uncertainty) efficiently. We observe significant performance improvement through our simultaneous learning framework.
>
> Nuria Armengol Urpi, Sebastian Curi, and Andreas Krause. Risk-averse offline reinforcement learning. ICLR, 2021
>
> Jiafei Lyu, Xiaoteng Ma, Xiu Li, and Zongqing Lu. Mildly conservative q-learning for offline reinforcement learning. NeurIPS, 2022.
>
> Yecheng Ma, Dinesh Jayaraman, and Osbert Bastani. Conservative offline distributional reinforcement learning. NeurIPS, 2021.
>
> > p2.par2 - ”We conducted extensive experiments in three different environments covering both risk-sensitive” - it’s 3 different benchmarks, you use multiple environments of D4RL.
>
> We appreciate the reviewer's comment. We apologize for the potential confusion of benchmarks and environments. In order to address this, we have revised the sentence to replace such ambiguous wordings to improve readability. We use three different benchmarks here: we conduct the experiments on risk-neutral D4RL (i.e., the traditional D4RL), risk-sensitive D4RL (i.e., the modified version of D4RL which includes the risk in the environment) and the risky robot navigation. We strictly/exactly follow the evaluation protocol in previous works (Ma et al., 2021, Urpi et al. 2021), and evaluate the algorithms on the same benchmarks.
>
> > Q: p3. par2 - "For simplify" - to simplify
>
> We appreciate the reviewer's comment. We have made the revision accordingly.
>
> >  In Section 4 and earlier you talk about "risk-distortion operators", but you do not define the concept in the background section, yet it seems essential for your method.
>
> We appreciate the reviewer's comment. We have added a footnote to provide the background knowledge of the risk-distortion operators. It is a class of distortion operators that also consider the risk. And we have added the relevant citation (Wang 2000) to improve the clarity.
>
> Shaun S Wang. A class of distortion operators for pricing financial and insurance risks. Journal of risk and insurance, pp. 15–36, 2000
>
> > In Section 4.1 you talk about implicit quantile functions, yet this concept has not been defined (what is implicit about them?) and no reference is given, I suppose you need [1].
>
> We appreciate the reviewer's comment. We have added the relevant citation [1] to the revised version,
>
> > There is no citation for the quantile Huber loss, you should provide a citation for it and refer to the equations the first time it is mentioned (ideally the definition would preceed the first usage of the function).
>
> We appreciate the reviewer's comment. We have added the citation of huber loss [Huber 1992].
> Peter J Huber. Robust estimation of a location parameter. In Breakthroughs in statistics: Methodology and distribution, pp. 492–518. Springer, 1992.

---

> ### Author Response · Authors · 2023-11-15
> **Response to Reviewer LtLP - Part 2**
>
> > p3, last par - you say that you use both uniform sampling and a quantile proposal network, then you say that you don't use a quantile proposal network and leave it for future work? If you don't use it, don't mention it here or mention it as a comment (an alternative is to use a quantile proposal network such as in (xyz citation)
>
> We value the reviewer's feedback. The intent behind the paragraph was to highlight two distinct methods for controlling the quantile level. We've revised this section to delineate the second approach as a comment, along with referencing relevant citations (Yang et al. 2019), clarifying that it stands as a potential alternative for future exploration.
>
> Derek Yang, Li Zhao, Zichuan Lin, Tao Qin, Jiang Bian, and Tie-Yan Liu. Fully parameterized quantile function for distributional reinforcement learning. NeurIPS 2019.
>
> >  In eq. 5, I suppose that you have N fixed quantile levels , hence the indexing. However, this was not mentioned previously, yet it is part of the methodology (also along the lines of earlier work?). Moreover, the loss is a function o and there is no expectation defined over s, a, s', a', though the quantile td error depends on the former. Also, "J" is commonly used as return? Also normalizing factor should be $N^2$?
>
> We appreciate the reviewer's consideration. Indeed, N represents the fixed quantile levels, although we regret not explicitly mentioning this earlier in the text, save for its inclusion in Algorithm 1. To address this, we've inserted a brief explanation regarding N following Equation 5. Additionally, we've revised Equation 5 to disambiguate the presence of two Ns, now denoted as N and K separately, indicating potential variations in quantile levels. Furthermore, to alleviate confusion, we've included the previously defined expectations over s, a, s', a' in Equation 5. In response to the concern about 'J,' we have removed it from the text to prevent any potential confusion.  We truly appreciate the reviewer raising the concern about the normalizing factor, we have made the adjustment accordingly based on the new N and K notations. We hope this revision can address your concerns.
>
> > eq. 6 - is it "distortion risk function" or "risk-distortion function"? I also don't understand this loss function, you have expectations wrt. marginal state distribution, however you have an action appearing in the implicit quantile function, where does the action come from? You are missing a policy here, please define this correctly.
>
> We thank the reviewer for the concerns. Addressing the query about the distortion risk function, we acknowledge and apologize for the confusion—it should indeed be termed the risk-distortion function, and we've rectified this in the paper. As for the policy clarification, we regret the oversight in representing a deterministic policy (i.e., $\pi_{\phi}(s) = a)$. To alleviate this confusion, we've updated the notation, replacing 'a' with $\pi_{\phi}(s)$. This revision aims to accurately reflect the involvement of the policy within the implicit quantile function. We hope this can address your concerns.
>
> > I think that your description of diffusion generative modelling needs significant adjustments. What you call an model is the denoiser, the Gaussian reverse process transition kernel arises from a continuous treatment of diffusion and exactly because of the choice of noise schedule. Since you don't provide contributions on level of diffusion generative modelling, this shouldn't take so much space of the paper.
>
> We sincerely value the reviewer's input. Section 4.2 primarily delineates why we opted for diffusion over cVAE and delves into the actor's components. The section discussing the diffusion model in the second part of Section 4.2 spans about half a page and includes four important formulas (8-11). These equations play a crucial role in facilitating the comprehension and replication of how RL is integrated into the diffusion process in our paper. Furthermore, the diffusion model employed in this study is not the standard DDPM; we have adapted the DDPM to create a conditional model, using states as conditions and actions as outputs. Consequently, our model can efficiently generate the behavior model in just a few steps (in our experiments, the number of steps is less than 3).
>
> > p6 - CQL is not a distributional RL algorithm?
>
> We are thankful for the reviewer's comment. CQL, despite not being a distributional algorithm, falls within the realm of offline RL algorithms. We included CQL as a baseline due to its widespread use in the field of offline RL as well as the recent advance in distributional RL algorithm, acknowledging its common usage as a benchmark in this context.
>
> >suggestion for table 1 - use colors instead of bold and *
>
> We truly value the concerns raised by the reviewer. In response, we've made adjustments by changing the formatting, switching bold text to red and asterisks (*) to blue, aiming to enhance the overall readability.

---

> > ### Comment · Reviewer_LtLP · 2023-11-21
> > **Reviewer response part 2**
> >
> > I thank the authors for the accepting some of the points that I stated, we are getting there.
> >
> > Regarding CQL, I am referring to the 1st paragraph on the page:
> > ```
> > . Moreover, ORAAC, CODAC and CQL are distributional RL algorithms where ORAAC and CODAC belong to offline RL.
> > ```
> > i.e., again, CQL is not a distributional RL algorithm.
> >
> >
> > > We sincerely value the reviewer's input. Section 4.2 primarily delineates why we opted for diffusion over cVAE and delves into the actor's components. The section discussing the diffusion model in the second part of Section 4.2 spans about half a page and includes four important formulas (8-11). These equations play a crucial role in facilitating the comprehension and replication of how RL is integrated into the diffusion process in our paper. Furthermore, the diffusion model employed in this study is not the standard DDPM; we have adapted the DDPM to create a conditional model, using states as conditions and actions as outputs. Consequently, our model can efficiently generate the behavior model in just a few steps (in our experiments, the number of steps is less than 3).
> >
> > I am still not convinced about your DDPM argument, the fact that the denoiser is conditioned on the state is not novel, also the part that it is jointly trained with the actor is just combining the diffusion objective and actor objective (whilst the diffusion objective is independent of the actor). I.e., in effect, you are just using a generative model (here, diffusion) to fit the conditional distribution of the behavior policy - am I saying something wrong here? The novelty that I see is the way that the actor is phrased (eq. 3) with the diffusion, however the perturbation phrasing is as well coming from previous work.
> >
> > Can you please state in clear terms what is the novelty of this paper in the general answer (so the way the diffusion is phrased is not, i.e. it's just using a generative model for modeling the behavior policy, the perturbation method of phrasing the actor is not (work from Fujimoto et al. used also by Urpi et al), the usage or risk distortion functions and distributional RL for risk-averse offline RL is not (Urpi et al.). Combination of the former?

---

> > > ### Author Response · Authors · 2023-11-22
> > >
> > > >Regarding CQL, I am referring to the 1st paragraph on the page: Moreover, ORAAC, CODAC and CQL are distributional RL algorithms where ORAAC and CODAC belong to offline RL.
> > >
> > > Thank you for your comments. We have revised this paragraph.
> > >
> > > >I am still not convinced about your DDPM argument, the fact that the denoiser is conditioned on the state is not novel, also the part that it is jointly trained with the actor is just combining the diffusion objective and actor objective (whilst the diffusion objective is independent of the actor). I.e., in effect, you are just using a generative model (here, diffusion) to fit the conditional distribution of the behavior policy - am I saying something wrong here? The novelty that I see is the way that the actor is phrased (eq. 3) with the diffusion, however the perturbation phrasing is as well coming from previous work. Can you please state in clear terms what is the novelty of this paper in the general answer (so the way the diffusion is phrased is not, i.e. it's just using a generative model for modeling the behavior policy, the perturbation method of phrasing the actor is not (work from Fujimoto et al. used also by Urpi et al), the usage or risk distortion functions and distributional RL for risk-averse offline RL is not (Urpi et al.). Combination of the former?
> > >
> > > We appreciate your concern about the novelty. We do agree with your understanding. We did not make a significant modification to the diffusion model, we phrased the actor to incorporate the diffusion. In general, in this work, we developed an innovative way to effectively integrate the diffusion model into the distributional offline RL framework to address risk-sensitive tasks. To our knowledge, we first (at least back to the time when submitting) introduce the diffusion model to this target domain (risk-sensitive offline RL). Empirically, we've got a good performance through the different benchmarks and we believe our work does provide useful inspiration and prove a promising direction for this research topic in the community.

---

> ### Author Response · Authors · 2023-11-15
> **Response to Reviewer LtLP - Part 3**
>
> >  figure 2 needs correction, y axis is not visible, also there are no error bands - how do we know what is the stability across reruns?
>
> We are greatly appreciating the reviewer's comments. In response, we have made revisions to Figure 2, introducing error bars to illustrate the stability. We believe that these updates in the new figure adequately address your concerns.
>
> >Is the diffusion generative model absolutely necessary in order to achieve this performance? Would you switch it out for a CVAE in order to compare?
>
> We genuinely appreciate the reviewer's concerns. Regarding the suggestion to replace the CVAE with the proposed diffusion generative model, in Section 4.2 on Page 4, we have provided a detailed discussion explaining why we opted for the diffusion method over the cVAE model. Moreover, we have also mentioned that the limitation of the cVAE in the introduction section paragraph 2 on Page 1. The key idea behind this is that the behavior policies on the offline dataset are normally collected from different agents and may have different modalities. It implies that some behavior policies may be sub-optimal. And recent works like Pearce et al., 2023 and Wang et al., 2023 also demonstrate that cVAE may not have a strong expressive ability than diffusion in offline RL. In order to further support our claim, we have provided the experiments here about replacing the diffusion model with cVAE. We hope this can address your concern.
>
> |              |           |                    |          |          |                    |           |          |
> |:------------:|:---------:|:------------------:|:--------:|:--------:|:------------------:|:---------:|:--------:|
> |              | Algorithm |       Medium       |          |          |       Expert       |           |          |
> |              |           | $CVaR_{0.1}$ |   Mean   | Duration | $CVaR_{0.1}$ |   Mean    | Duration |
> | Half-Cheetah | UDAC-CVaE |      221(40)       | 346(54)  |  200(0)  |      633(158)      | 1205(87)  |  200(0)  |
> |              |  UDAC-D   |      276(29)       | 417(18)  |  200(0)  |      732(104)      | 1352(100) |  200(0)  |
> |  Walker-2D   | UDAC-CVaE |      689(144)      | 1255(88) | 395(45)  |      1277(84)      | 1988(69)  | 445(18)  |
> |              |  UDAC-D   |      1368(89)      | 1602(85) | 406(24)  |      1398(67)      | 2198(78)  | 462(20)  |
> |    Hopper    | UDAC-CVaE |      1425(44)      | 1502(17) | 488(10)  |      1022(45)      | 1405(50)  |  492(9)  |
> |              |  UDAC-D   |      1502(19)      | 1602(18) |  480(6)  |      1302(28)      | 1502(27)  |  494(8)  |
>
> Tim Pearce, Tabish Rashid, Anssi Kanervisto, Dave Bignell, Mingfei Sun, Raluca Georgescu, Sergio Valcarcel Macua, Shan Zheng Tan, Ida Momennejad, Katja Hofmann, and Sam Devlin. Imitating human behaviour with diffusion models. ICLR, 2023
>
> Zhendong Wang, Jonathan J Hunt, and Mingyuan Zhou. Diffusion policies as an expressive policy class for offline reinforcement learning. ICLR, 2023

---

> > ### Comment · Reviewer_LtLP · 2023-11-21
> > **Reviewer response 3**
> >
> > Thank you for providing the result table for CVAE, this answers my question partially. Is the likelihood of the data in the case of the CVAE lower than in the case of the diffusion model? You mean to say that the CVAE can't fit the distribution of the data?

---

> ### Author Response · Authors · 2023-11-22
>
> >Thank you for providing the result table for CVAE, this answers my question partially. Is the likelihood of the data in the case of the CVAE lower than in the case of the diffusion model? You mean to say that the CVAE can't fit the distribution of the data?
>
>
>
> We appreciate the reviewer's concern. We agree that it would be possible that the likelihood of the data in this case of the cVAE is lower than the diffusion model. Based on our existing experiments, which reflect that the cVAE may not be able to perform as well as diffusion. However, we believe that more extensive experiments and theoretical analysis should be carried out to further investigate this problem. We do appreciate the reviewer's point and will target this as future work.
>
> We can not certainly say that the cVAE can't fit the distribution of the data, the possible justification here could be that: the cVAE may not have a strong expressive as what the diffusion model has.

---

> ### Author Response · Authors · 2023-11-23
> **Looking forward your feedback**
>
> Dear reviewer LtLP,
>
> Thank you again for your valuable comments on our paper. We hope that our recent response can address your additional concerns.  We are mindful that you may be currently facing a demanding schedule and truly appreciate the time and effort you invested in reviewing our response. As the ICLR discussion deadline approaches (9 hours remaining), we kindly request your feedback on our submitted response. Your comments are crucial to us, and we understand that this may be the final opportunity for us to address any additional concerns you might have.
>
> Thank you once again for your time and engagement.
>
> Best regards,
> The Authors

---

> > ### Comment · Reviewer_LtLP · 2023-11-23
> > **Reviewer response**
> >
> > I thank you for engaging with me during the discussion phase and providing additional results regarding the CVAE vs diffusion issue. Since you made updates to the paper and ran the CVAE as replacement to the diffusion model, I am inclined to raise my score by 1. However, since really the modification that you propose is only a tiny fraction of the paper (the diffusion model behavior policy), the analysis why this diffusion model is better than any other generative model for the case of this type of data is not there (except supposedly reflected in the performance of the policy).
> >
> > For a conference like ICLR, I believe that there is a need for some more fundamental results. However, I need to clear this up with the area chair and the other reviewers who gave a positive score to your paper.

---

> > > ### Author Response · Authors · 2023-11-23
> > >
> > > Thank you for your justification. For your reference, we have also conducted experiments in risky environments (i.e., Risky robot navigation) to support the proposed method. We hope these additional results could also provide insights into how our model performs in risky environments other than MuJoCo. Given that our goal is to learn an uncertainty-aware policy, we did not perform experiments in risk-neutral environments. We trust these additional experiments can address your concerns (fully or partially) about the different types of data.
> > >
> > > Moreover, we would like to clarify that the reason we only conducted experiments on Half-Cheetah, Walker-2D, and Hopper is that we are following the evaluation process outlined in previous works (Urpi et al, Ma et al.).
> > >
> > > |           |                 |           |                    |            |              |              |                    |             |
> > > |:---------:|:---------------:|:---------:|:------------------:|:----------:|:------------:|:------------:|:------------------:|:-----------:|
> > > | Algorithm | Risky PointMass |           |                    |            |  Risky Ant   |              |                    |             |
> > > |           |      Mean       |  Median   | $CVaR_{0.1}$ | Violations |     Mean     |    Median    | $CVaR_0.1$ | Violations  |
> > > |  UDAC-D   |    -5.8(1.4)    | -4.2(1.3) |     -12.4(2.0)     |    0(0)    | -392.5(33.2) | -399.5(40.5) |    -598.5(50.2)    | 291.5(20.8) |
> > > | UDAC-cVAE |    -6.3(1.3)    | -5.2(1.7) |     -15.9(1.2)     |    0(0)    | -402.5(28.2) | -410.3(37.6) |    -589.3(66.1)    | 321.8(32.1) |

---

### Author Response · Authors · 2023-11-21
**Thanks for Reviewers' Meaningful Comments**

Dear Reviewers,

As we are approaching the end of the rebuttal phase. May we kindly request that you take a moment to review our response to your comments? If there are any aspects that require further clarification or if you have additional questions, please let us know so that we can respond ASAP. Additionally, if you find our response satisfactory, we would greatly appreciate it if you could consider increasing the rating you have given us. Your feedback is invaluable to us and helps us continually improve our paper.

Thanks very much.

---

### Meta-Review · Area_Chair_NrTs · 2023-12-12

**Metareview:**

### Summary
This paper introduces a new model-free offline RL algorithm that simultaneously addresses epistemic and aleatoric uncertainties. The algorithm utilizes a distributional RL framework to model the distribution of discounted cumulative rewards and a diffusion model to capture the behavior policy. Extensive experiments in risk-sensitive and risk-neutral environments demonstrate the algorithm's superior performance compared to existing offline RL methods.

### Decision
This paper addresses a significant problem in offline RL with an interesting approach to addressing uncertainty quantification. The proposed approach is reasonable, and the experimental results are convincing. However, the writing of this paper had some problems, as pointed out by Reviewer LtLP. Due to the lack of clarity, some of the points made by the paper were missed. Ultimately, none of the reviewers were willing to champion this paper for acceptance. As it stands, this paper feels a bit rushed for publication. I recommend the authors work on the clarity of the paper, address all the issues pointed out by the reviewers, and submit it to a different venue.

**Justification For Why Not Higher Score:**

This paper was borderline. It feels like, as it stands, the paper feels rushed, and I do believe that it can benefit from another iteration of reviews after a rewrite.

**Justification For Why Not Lower Score:**

N/A

---

### Decision · Program_Chairs · 2024-01-16

Reject